# NDRGs in Breast Cancer: A Review and In Silico Analysis

**DOI:** 10.3390/cancers16071342

**Published:** 2024-03-29

**Authors:** Emilly S. Villodre, Anh P. N. Nguyen, Bisrat G. Debeb

**Affiliations:** 1Department of Breast Medical Oncology, The University of Texas MD Anderson Cancer Center, Houston, TX 77030, USA; esschlee@mdanderson.org (E.S.V.); anhpnnguyen3@gmail.com (A.P.N.N.); 2MD Anderson Morgan Welch Inflammatory Breast Cancer Clinic and Research Program, The University of Texas MD Anderson Cancer Center, Houston, TX 77030, USA

**Keywords:** breast cancer, NDRGs, NDRG1, NDRG family, tumor promoter, tumor suppressor

## Abstract

**Simple Summary:**

Breast cancer is the most common cancer in women and one of the deadliest. While survival rates for patients with breast cancer have seen notable improvements in recent decades, current treatment strategies still face significant limitations, especially for patients with aggressive, therapy-resistant, and metastatic breast cancers. For this reason, it is important to identify new targets in order to develop more effective therapeutic strategies. The N-myc downstream regulated gene family (NDRGs), comprising *NDRG1*, *NDRG2*, *NDRG3*, and *NDRG4*, has been previously described as tumor suppressors. However, recent findings challenge this perception, particularly for *NDRG1*, which has demonstrated a critical role in driving tumor growth and metastasis in aggressive forms of breast cancer. In this review, we discuss the role of the NDRG family members in breast cancer, which is supported by analyses of genomic and transcriptomic data from various independent breast cancer patient cohorts.

**Abstract:**

The N-myc downstream regulated gene family (NDRGs) includes four members: *NDRG1*, *NDRG2*, *NDRG3*, and *NDRG4*. These members exhibit 53–65% amino acid identity. The role of NDRGs in tumor growth and metastasis appears to be tumor- and context-dependent. While many studies have reported that these family members have tumor suppressive roles, recent studies have demonstrated that NDRGs, particularly *NDRG1* and *NDRG2*, function as oncogenes, promoting tumor growth and metastasis. Additionally, NDRGs are involved in regulating different signaling pathways and exhibit diverse cellular functions in breast cancers. In this review, we comprehensively outline the oncogenic and tumor suppressor roles of the NDRG family members in breast cancer, examining evidence from in vitro and in vivo breast cancer models as well as tumor tissues from breast cancer patients. We also present analyses of publicly available genomic and transcriptomic data from multiple independent cohorts of breast cancer patients.

## 1. Introduction

Breast cancer is the most common cancer and the leading cause of cancer-related deaths among women in the United States. The survival rate for patients with breast cancer has increased over the past few decades, in part because of improved early diagnosis and advanced treatments such as targeted and hormonal therapies [1,2]. Breast cancer can be classified in two ways. One classification, molecular subtyping, considers gene expression data and groups breast cancer into four main categories: luminal A, luminal B, HER2, and basal-like. A fifth subtype, normal-like, has also been described, although concern has been expressed that it may be an artifact [3,4,5,6]. The other classification is the one currently used in the clinic for subtyping and treatment decisions. Protein expression patterns are first identified by the pathologist using immunohistochemical staining for the estrogen receptor (ER), progesterone receptor (PR), and human epidermal growth factor receptor 2 (HER2); Ki67, a marker of proliferation, was added later. These four markers are used to create four surrogate intrinsic subtypes: luminal A (ER^+^, PR^+^, HER2^−^, Ki67-low), luminal B (HER2^−^ [ER^+^, PR^−^, HER2^−^, Ki67-high), HER2^+^ (ER^+^, PR^+/−^, HER2^+^, Ki67-low/high), HER2 enriched (ER^−^, PR^−^, HER2^+^, Ki67-high), and triple-negative (ER^−^, PR^−^, HER2^−^, Ki67-high) (Figure 1) [7,8,9,10]. Notably, triple-negative breast cancer is not synonymous with basal-like breast cancer, although they do share some similarities. According to Bertucci and colleagues, 71% of triple-negative breast cancers are basal-like, and 77% of basal-like breast cancers are triple-negative [11,12,13].

Breast cancer treatment is currently based on the disease stage and clinical tumor subtype. In early stages, breast cancer can be curable by mastectomy, radiotherapy, endocrine therapy, chemotherapy, and bone-stabilizing drugs [2]. With the rapid development of genetic tools and high-throughput techniques, as well as the increased understanding of the molecular events involved in cancer development and progression, scientists have focused on identifying novel therapeutic targets to improve survival outcomes for patients with solid tumors. In breast cancer, anti-HER2–targeted therapies such as trastuzumab have been effective against HER2^+^ breast cancer [14,15,16,17]. Other targets showing promise in preclinical breast cancer models include epidermal growth factor inhibitor (EGFR) and vascular endothelial growth factor (VEGF) [18,19,20]. Because existing breast cancer treatment strategies are not fully curative, the search for novel targeted therapies continues, with the goal of developing new, more effective therapeutics, particularly for aggressive, therapy-resistant, and metastatic breast cancers.

The origin of the N-myc downstream regulated gene (NDRG) family members, their biological roles during development, and their function in various cancer types are reviewed elsewhere [21,22,23,24]. Subsequent reviews have focused on the literature analyses of the roles of NDRGs in different cancer types and have highlighted their importance during hypoxia [24] and their role in gut development [21] or central and peripheral nervous system development [22]. Associations between NDRGs and tumorigenesis have been reported, with most reports focusing on their role as tumor suppressors. NDRG1 is described as a suppressor in prostate, ovarian and colorectal cancer, while in gastric, hepatocellular carcinoma and bladder cancer it was found to be a promoter. NDRG2 is mainly a suppressor in ovarian, renal, and glioblastoma cancers, among others, while NDRG3 has been described as a promoter in gastric and hepatocellular carcinoma. Meanwhile, NDRG4 is used as an early detection marker in colorectal cancer. However, accumulating evidence indicates that the presence of NDRGs, particularly NDRG1, is negatively correlated with patient prognosis and is associated with increased tumor progression and metastasis in various types of cancer [25,26,27].

This review summarizes the current literature on the roles of NDRGs in prognosis, tumor progression, and metastasis in breast cancer models. We also present our in silico analyses of NDRGs based on the available genomic and transcriptomic data from several independent cohorts of patients with breast cancer. In general, we found that NDRG2, -3, and -4 were not statistically correlated with outcomes, but high NDRG1 expression correlated with worse outcomes. Notably, *NDRG1* is highly amplified in breast cancer, particularly in basal-like and metastatic breast cancers.

## 2. NDRG Family Proteins

To date, four types of NDRG proteins have been discovered: NDRG1, NDRG2, NDRG3, and NDRG4; these proteins exhibit 53–65% amino acid identity [28]. These NDRGs are predominantly located in the cytosol and share the α/β hydrolase fold region, although their hydrolase catalytic activity has not been elucidated [23,24]. NDRG1 can also be found in the nucleus and cell membrane. The genes for NDRG proteins are also highly conserved in a variety of species, underscoring their critical roles in cell development [21,28]. During evolution, the functions of the four genes in the NDRG family have diverged after numerous duplication events to form four separate homology clusters. According to the phylogenetic tree, *NDRG1* and *NDRG3* are the most closely related to each other with 67% homology, whereas *NDRG2* is more closely related to the ancestral gene in the phylogenetic tree than to the other NDRGs [23]. Phylogenetic analysis has also indicated that human *NDRG1* and *NDRG2* are most closely related to the homologous genes in macaques. Meanwhile, the branches of human *NDRG3* and *NDRG4* are closer to rodent homologous genes than to other species [23].

Human *NDRG1* was first described by Ulrix et al. in 1999; Qu and colleagues were the first to describe other NDRGs in humans [28,29]. *NDRG1*, *NDRG2*, *NDRG3*, and *NDRG4* are located on chromosomes 8q24.22, 14q11.2, 20q11.23, and 16q21, respectively [30] (Figure 2). The encoded protein of NDRG1 (also known as CAP43 or DRG1) has 394 amino acids and a molecular mass of 43 kDa [30]. This gene has important roles in cell proliferation, stress response, and inflammatory processes such as allergy and wound healing [23]. NDRG2 is a 41 kDa protein composed of 371 amino acids [30]. NDRG2 has been reported to be a tumor suppressor and to inhibit glucose uptake in breast cancer cells [31]. NDRG3 has 375 amino acids and a molecular mass of 41.5 kDa; it is highly expressed in the testis, prostate, and ovary [30,32]. NDRG4 encodes a 352-amino acid protein with a molecular mass of 40 kDa [30]. NDRG4 has been negatively correlated with outcomes in different cancer types, and its methylation has served as a biomarker in colorectal and gastric cancer [33,34]. Limited information is available regarding post-translational modifications of the NDRG family members. The most extensively described is the different phosphorylation sites of NDRG1, mediated by SGK1 and GSK3β. However, small ubiquitin-like modifier (SUMO) modification and/or cleavage of NDRG1 have also been reported [35,36].

## 3. NDRGs and Breast Cancer

The current literature on the function of NDRGs in breast cancer is summarized in Table 1. Details of each family member are given briefly below.

### 3.1. NDRG1

Although NDRG1 was first considered to suppress metastasis in breast cancer, evidence is mounting to indicate that NDRG1 has pro-oncogenic and pro-metastatic effects in breast cancer. Bandyopadhyay and colleagues were the first to show that NDRG1 overexpression in breast cancer cell lines inhibited invasion in vitro, and that NDRG1 expression was regulated by PTEN [37,38]. Fotovati et al. have analyzed several breast cancer cell lines and found that both the expression and inhibition of NDRG1 induced by estradiol depended on the estrogen receptor status of the cell lines [39]. Other studies have concluded that depletion of NDRG1 promotes proliferation and increases migration and invasion [40,41,42,43,44,45,46]. Redmond and colleagues have shown that TBX-2 acts as a tumor promoter in MCF-7 breast cancer cells by inhibiting NDRG1 expression and inducing proliferation; Chiang et al. have reported that NDRG1 suppresses the tumor promoter function of WISP1 by suppressing its effects on the proliferation and invasion of MCF-7 and MDA-MB-231 breast cancer cells in vitro [42,46]. Salis et al. have observed that the effects of NDRG1 on the in vitro migration of MCF-7 breast cancer cells are regulated by the TGF-β pathway: NDRG1 mRNA expression levels were reduced after treatment with TGF-β1, but treatment with the cholesterol-reducing agent fluvastatin inhibited the migration of the TGF-β1-treated cells and increased NDRG1 expression [41]. Silencing SGK1, a kinase known to regulate NDRG1 phosphorylation, was also found to inhibit NDRG1 expression and increase breast cancer cell migration and invasion in vitro [45,47]. NDRG1 was also correlated with the expression of proteins associated with epithelial traits (high E-cadherin, low vimentin expression) [48]. More recently, Abascal and colleagues established that NDRG1 expression depended on which progesterone receptor isoform was expressed by T47D cells, with higher NDRG1 in progesterone B cells than in progesterone A cells [49]. Interestingly, NDRG1 expression is also involved in the differentiation of breast cancer cells in vitro and in vivo: overexpression of NDRG1 in MDA-MB-231 cells led to increased areas of well-differentiated tumors in xenograft mouse models [50]. However, none of these studies have included in vivo experiments to confirm the in vitro findings on the proliferation and migration/invasion of NDRG1-high vs. NDRG1-low cells, and only two studies have reported outcomes from patients with breast cancer. One of those two studies has found that breast cancer patients with NDRG1-negative tumors had worse disease-free survival than patients with NDRG1-positive tumors [38]. In the other study, analysis of NDRG1 and PTEN regarding outcomes revealed that patients with PTEN-negative and NDRG1-positive tumors had better outcomes than the other groups (PTEN^−^ and NDRG1^−^; PTEN^+^ and NDRG1^−^; PTEN^+^ and NDRG1^+^) [37].

Contrary to the aforementioned studies, a substantial body of research consistently demonstrates that NDRG1 serves as a marker of poor prognosis in patients with breast cancer and functions as a promoter of tumor progression and metastasis in breast cancer models [51,52,53,54,55,56,57,58,59,60,61]. Mao and colleagues have shown that NDRG1 was associated with the progression of breast cancer from atypia to carcinoma, in that its expression was mostly negative in normal epithelium, atypical hyperplasia, and carcinoma in situ but was strongly positive in invasive carcinoma, especially invasive ductal carcinoma (92% of the cases) [58]. In terms of survival outcomes, Nagai et al. have analyzed nearly 600 samples from patients with breast cancer and found that NDRG1^+^ tumors were associated with worse outcomes (overall and disease-free survival) than NDRG1^−^ tumors, with 10-year survival rates of 34% for NDRG1^+^ tumors vs. 67% for NDRG1^−^ tumors [54]. In a meta-analysis of 23 breast cancer cohorts and more than 3500 samples, Sevinsky et al. have shown that high NDRG1 expression was correlated with worse recurrence-free and metastasis-free survival [54]. Similarly, in our own study of patients with inflammatory breast cancer, a rare and highly aggressive variant of breast cancer, we found that tumors with high NDRG1 expression were associated with worse overall and disease-specific survival compared with NDRG1-low tumors. In the same cohort, NDRG1 was found to be an independent prognostic factor for worse outcome [52]. Consistent with these observations, we further found in a later analysis of 216 patients with all breast cancer subtypes that high NDRG1 expression was associated with aggressive breast tumor features (i.e., ER^−^ status, triple-negative disease, and high grade) and was also independently associated with poor outcome [51]. In a meta-analysis of more than 3000 patients with breast cancer, de Nonneville et al. also found that NDRG1 was an independent predictor of overall survival in patients with ER^+^/HER2^−^ breast cancer: tumors with high NDRG1 expression were associated with worse overall survival at 10 years compared with NDRG1-low tumors [60]. A recent comprehensive phosphoproteomic profiling study of breast cancer tumors also identified NDRG1 among the top genes associated with basal-like intrinsic breast cancer subtypes; both PAM50 and non-negative matrix factorization clustering analyses linked NDRG1 with aggressive breast tumor features [62]. López-Tejada and colleagues have observed similar results in their analysis of 83 samples from breast cancer patients, showing that high NDRG1 was associated with poor cumulative survival [63]. That group also found that negative nuclear phospho-NDRG1 was associated with poor cumulative survival. Collectively, these findings show that NDRG1 is correlated with aggressiveness and poor survival outcomes in patients with breast cancer.

In support of these patient-sample analyses, others have described NDRG1 as having pro-oncogenic functions in models of aggressive breast cancer. Specifically, depletion of NDRG1 was shown to inhibit proliferation, invasion, migration, and cancer stem cell subpopulations in vitro [51,54,57,59]. Notably, NDRG1 expression was increased in breast cancer cell lines that are resistant to AKT inhibitors [53], indicating that NDRG1 levels could be associated with responsiveness to available treatments. Our own in vivo studies using an immunocompromised mouse model revealed that depletion of NDRG1 in aggressive breast cancer cell lines significantly delayed tumor latency and reduced the size of the primary tumor; of particular note, we found that tail-vein injection of NDRG1-depleted breast cancer cell lines into immunocompromised mice suppressed metastatic burden and the incidence of brain metastasis and extended survival relative to NDRG1-expressing breast cancer cells [51]. Berghoff et al. have similarly found that high-NDRG1-expressing, slow-cycling breast cancer cells exhibited cancer stem cell features and were responsible for the development of brain metastases [61]. These in vitro and in vivo findings provide strong evidence that NDRG1 plays a significant role in promoting the metastasis of aggressive breast cancer cells.

### 3.2. NDRG2

The function of NDRG2 in breast cancer has not been studied as extensively as NDRG1. Although most studies have concluded that NDRG2 acts as a tumor suppressor in breast cancer [26,64,65,66,67,68,69,70,71,72,73,74] (Table 1), a single study has described NDRG2 as having both suppressor and promoter functions depending on the subtype of breast cancer [75].

A group led by Jong-Seok Lim has published many articles on the function of NDRG2 in breast cancer; they have described how overexpression of NDRG2 led to a decreased proliferation, survival, and migration and invasion potential of breast cancer cells in vitro through the regulation of various signaling pathways [66,67,71,72,73,74]. Park et al. have found that the overexpression of NDRG2 in MDA-MB-231 cells induced the expression of SOCS1, which in turn inhibited JAK2/STAT3 signaling [67]. In other studies, overexpressing NDRG2 in MDA-MB-231 cells reduced their migration and invasion potential in vitro through the induction of BMP-4, which suppressed the activity of MMP-9 [66], and by downregulating COX-2 through NF-kB signaling [72]. Indeed, those NDRG2-overexpressing MDA-MB-231 cells induced apoptosis and inhibited the epithelial–mesenchymal transition, the latter by reducing STAT3/Snail signaling [71,73]. Interestingly, breast cancer cells expressing NDRG2 have also inhibited osteoclast differentiation by downregulating secreted ICAM1 expression, suggesting that NDRG2 may suppress bone metastases from breast cancer [74]. However, these reports were all limited to in vitro studies of breast cancer cell lines.

That said, others have found results that support the aforementioned findings in analyses of breast cancer patient samples and xenograft mouse models [68,70,76,77]. Liu et al. have analyzed mechanisms involved in downregulation of NDRG2 in 13 breast cancer cell lines and 21 paired breast cancer–normal tissue samples from patients, and they found that NDRG2 mRNA and protein levels were reduced in eight cell lines and five breast cancer samples, with the mechanism of downregulation being complex and dependent on cell type [26]. Lorentzen et al. have also found that NDRG2 mRNA expression was reduced in breast cancer samples compared with normal tissue samples [69]. Two other analyses of large groups of patients with breast cancer have shown that patients with low-NDRG2-expressing tumors had worse disease-free survival [68] or worse overall survival [64].

On the other hand, another group have found NDRG2 to have a pro-oncogenic role in breast cancer; breast tumor samples had lower expressions of NDRG2 than normal breast tissue, and basal-like tumors had higher NDRG2 expression levels compared with luminal tumors. Further analysis of samples from 211 breast cancer patients has revealed a positive correlation between high NDRG2 expression and worse outcome for patients with basal-like tumors, as well as favorable overall and relapse-free survival for patients with luminal A breast cancer, supporting the contention that NDRG2 is associated with aggressiveness and unfavorable outcomes in aggressive breast cancers [75]. Interestingly, this group have found further differences in the expression and methylation patterns of NDRG2 between basal and luminal breast cancers, with the NDRG2 tumors having lower methylation and increased expression compared to the luminal tumors. Consistent with these findings from patient samples were in vitro findings that silencing NDRG2 reduced proliferation and migration in basal-like A HCC1806 cells, whereas overexpressing NDRG2 in luminal-type MCF7 cells reduced proliferation [75]. These observations support the concept that the roles of NDRG2 in breast cancer differ according to molecular subtype.

### 3.3. NDRG3

Much less is known about the role of NDRG3 in breast cancer, with only two studies published to date; one of those studies has reported a potential tumor suppressor function and the other a tumor promoter role for NDRG3 [78,79]. In the first of these studies, Estiar and colleagues analyzed tissue samples from 88 patients and found that NDRG3 was downregulated in patients with breast cancer, and that patients with a low tumor expression of NDRG3 had worse outcomes than those with a high NDRG3-expressing tumors [79]. However, the other study, which involved samples from 1339 patients with invasive breast cancer, concluded NDRG3′s potential tumor promoter role in breast cancer: patients with NDRG3^+^ tumors had worse overall survival than patients with NDRG3– tumors, and NDRG3 independently predicted worse overall and disease-free survival [78].

### 3.4. NDRG4

The only study published to date on NDRG4 indicates that NDRG4 may function as a tumor suppressor in breast cancer. An Analysis of samples from breast cancer patients showed that NDRG4 was highly methylated, and patients with methylated NDRG4 had worse overall and distant metastasis-free survival relative to those with non-methylated NDRG4 tumors. Moreover, NDRG4 methylation status was an independent prognostic factor for distant metastasis-free survival [80].

**Table 1 cancers-16-01342-t001:** Summary of known functions of NDGRs in breast cancer.

	Function	References	Cell Lines Analyzed	No. of Patients Analyzed	Experimental Conditions		Major Conclusions
					In Vitro	In Vivo	
**NDRG1**	**Tumor suppressor**	Bandyopadhyay, 2004 [38]	MDA-MB-468, MDA-MB-435, MDA-MB-231, MCF-7	85	Yes	No	1. Overexpression of NDRG1 reduced invasion of MDA-468 breast cancer cells in vitro.
							2. Treatment with 5-azacytidine suppressed breast cancer cell invasion in vitro.
							3. Patients with *NDRG1*-negative tumors had worse disease-free survival than those with *NDRG1* -positive tumors.
		Bandyopadhyay, 2004 [37]	MDA-468, BT-549	85	Yes	No	1. PTEN regulated NDRG1 expression.
							2. Patients with *PTEN*-negative and *NDRG1*-negative tumors had better outcomes than patients with PTEN-negative and NDRG1-positive tumors.
		Fotovati, 2006 [39]	SK-BR-3, MDA-MB-231, T47D, MCF-7, ZR75-1, R-27 (MCF-7	96	Yes	No	1. Cells that were resistant to tamoxifen expressed higher levels of NDRG1 than parental cells.
			tamoxifen-resistant cell line)				2. *NDRG1* expression was inversely correlated with estrogen receptor alpha (ER-α) expression.
							3. Treatment with 17β-estradiol E_2_ reduced NDRG1 levels in ERα-positive cell lines, but not in ERα-negative cell lines.
		Redmond, 2010 [42]	MCF-7, BT474, MDA-MB-157, MDA-MB-453,	Not reported	Yes	No	1. NDRG1 expression was repressed by TBX2 through EGR1.
			MDA-MB-231, MDA-MB-468, T47D, ZR75-1				2. TBX2 had a tumor promoter function and inhibited NDRG1 to promote cell growth.
		Fotovati, 2011 [50]	SK-BR-3, MDA-MB-231, T47D, MCF-7	45	Yes	Yes	1. NDRG1 expression was upregulated during the differentiation of breast cancer cells in vitro and could be used as a marker for differentiation of breast cancer.
							2. Induction of NDRG1 could be a strategy for cancer treatment.
		Lai, 2011 [43]	MCF-7	Not reported	Yes	No	1. NDRG1 was highly expressed upon reoxygenation, and reoxygenated cells showed increased levels of migration.
		Liu, 2011 [48]	MCF-7	33	Yes	Yes ^[a]^	^[a]^ Used WB1-1 cells (cell line isolated tumor cells from the mammary tumor of MMTV-Wnt mouse model).
							1. NDRG1 repressed Wnt-β pathway by interacting with LRP6 (Wnt co-receptor) and re-activating GSK3β.
							2. NDRG1 correlated with epithelial traits in breast cancer cell lines (high E-cadherin, low vimentin).
							3. NDRG1 functioned as a metastasis suppressor by inhibiting Wnt signaling.
		Han, 2013 [44]	MDA-MB-231, T47D	389	Yes	No	1. NDRG1 methylation status could be involved in tumorigenesis in breast cancer.
		Chiang, 2015 [46]	MCF-7, MDA-MB-231	Not reported	Yes	Yes ^[b]^	^[b]^ In vivo studies are focused on WISP1 not NDRG1
							1. Overexpression of WISP1 induced epithelial-mesenchymal–transition, migration, and invasion.
							2. NDRG1 expression was inhibited in WISP1-overexpressing cells.
							3. Overexpression of NDRG1 reduced the effect of WISP1 in proliferation and invasion of breast cancer cells.
		Salis, 2016 [41]	MCF-7	Not reported	Yes	No	1. *NDRG1* mRNA expression levels were reduced after treatment with TGF-β1.
							2. Treatment with fluvastatin inhibited migration of TGF-β1-treated cells and induced an increase in NDRG1 expression.
		Tian, 2017 [40]	MDA-MB-231, MDA-MB-453, MCF-7	Not reported	Yes	No	1. siNDRG1 promoted migration/invasion of MDA-231 breast cancer cells, an effect that was inhibited by treatment with an SGK1 inhibitor.
		Godbole, 2018 [45]	T47D, BT474, MDA-MB-231, ZR-75-1, MCF-7	Not reported	Yes	No	1. NDRG1 expression was inhibited after knocking down SGK1, increasing cell migration and invasion.
							2. Silencing of NDRG1 increased levels of phosphorylated EGFR (pEGFR), AKT (pAKT), and ERK1/2 (pERK1/2) in T47D and MDA-MB-231 cells.
		Abascal, 2022 [49]	T47D, MDA-231	Not reported	Yes	No	1. *NDRG1* expression is higher in Luminal B than in Luminal A tumors
							2. Metastatic potential of breast cancer was influenced by progesterone receptor isoforms (A or B) regulating NDRG1
	**Tumor promoter**	Nagai, 2011 [56]	Not reported	596	No	No	1. *NDRG1*-positive tumors were associated with worse outcomes.
							2. The 10-year overall survival rates were 67% for NDRG1-negative versus 34% for NDRG1-positive tumors.
		Mao, 2011 [58]	Not reported	215 + 20 ^[c]^	No	No	^[c]^ Involved 215 samples of different subtypes of breast cancer; 20 tumor tissues had a paired non-tumor portion.
							1. *NDRG1* expression was associated with a progression of breast cancer, from atypia to carcinoma development.
							2. *NDRG1* expression correlated with a high tumor category in invasive breast cancer.
		Sommer, 2013 [53]	BT-474, CAMA-1, ZR-75-1, T47D, HCC-1187, SUM-52-PE,	Not reported	Yes	No	1. NDRG1 and SGK1 expression were increased in AKT inhibitor-resistant cell lines.
			HCC-1937, MDA-MB-436, BT-549, MDA-MB-157,				2. High levels of SGK1 were one means of predicting resistance to AKT inhibitors.
			MDA-MB-231, HCC-1806, JIMT-1				3. Levels of phosphorylated NDRG1 could serve as a marker for response to AKT inhibitors.
		Parris, 2014 [55]	Not reported	229	No	No	1. *NDRG1* was hypomethylated and highly expressed in breast cancer samples.
		Li, 2016 [59]	MCF-7	Not reported	Yes	No	1. NDRG1 knockdown inhibited proliferation and migration; it induced cell cycle arrest under hypoxia.
		Sevinsky, 2018 [54]	SKBR3, MCF-7, HCC1569, BT474, MDA-MB-231,	3554 ^[d]^	Yes	No	^[d]^ Meta-analysis of 23 distinct breast cancer cohorts.
			MDA-MB-468				1. Patients with *NDRG1*-high tumors had worse recurrence- and metastasis-free survival.
							2. High expression of NDRG1 correlated with hypoxia and glycolytic pathways.
							3. NDRG1 knockdown reduced proliferation and led to dysfunction in lipid metabolism.
		Mishra, 2020 [57]	MDA-231, SUM159	Not reported	Yes	No	1. NDRG1 expression reduced in cybrids with benign mitochondria.
							2. NDRG1 knockdown reduced proliferation of SUM159 cells.
		Villodre, 2020 [51]	Not reported	64	No	No	1. *NDRG1* was an independent predictor of worse outcomes in inflammatory breast cancer.
							2. NDRG1, together with estrogen receptor status and disease stage, could be used to further stratify patient outcomes.
		Berghoff, 2021 [61]	JIMT1, MDA-231	74 + 61 ^[e]^	Yes	Yes	^[e]^ Involved 75 primary breast cancer and 61 breast cancer brain metastasis specimens.
							1. Slow-cycling cells efficiently formed brain metastasis and extracranial metastasis and expressed high levels of NDRG1.
							2. Silencing NDRG1 reduced the ability of cells to develop brain metastasis.
							3. Patients with high NDRG1-expressing tumors had worse metastasis-free survival.
		Villodre, 2022 [52]	SUM149, BCX010, MDA-IBC3	216	Yes	Yes	1. NDRG1 knockdown inhibited migration, invasion, and cancer-stem cell features in aggressive breast cancer cell lines.
							2. Silencing of NDRG1 inhibited primary tumor growth and brain metastasis.
							3. Patents with breast cancer and high NDRG1 expression had worse outcomes, and *NDRG1* was an independent prognostic factor.
		de Nonneville, 2022 [60]	Not reported	7850 ^[f]^	Not reported	Not reported	^[f]^ Involved 5929 ER+/HER2- and 1936 TN cases.
							1. Patients with *NDRG1* -high tumors had worse overall survival.
							2. The 10-year overall survival rates were 68% for NDRG1-high tumors versus 78% for NDRG1-low tumors.
							3. High expression of *NDRG1* was associated with aggressive tumor features.
							4. *NDRG1* was an independent predictor of overall survival in patients with ER+/HER2- disease.
		López-Tejada, 2023 [63]	BT549, Hs578T, MDA-MB-231, MDA-MB-436, MDA-MB-468, SUM159	83	Yes	No	1. High NDRG1 expression was associated with poor cumulative survival.
							2. Negative nuclear phospho-NDRG1 expression was associated with poor cumulative survival.
							3. Cellular expression and subcellular localization of NDRG1 and phospho-NDRG1 in TNBC correlated with patient survival.
							4. TGFβ governed the activity of NDRG1 in tumor progression to modulate epithelial–mesenchymal transition, metastasis, and the tumor-initiating capacity of cancer cells.
**NDRG2**	**Tumor suppressor**	Liu, 2007 [26]	MCF-7, MDA-MB-231, SK-BR-3	21	Yes	No	1. Low NDRG2 levels were observed in breast cancer cell lines and in 5 of 21 breast cancer tissues samples.
		Park, 2007 [67]	T47D, MCF-7, MDA-MB-453, MDA-MB-231	Not reported	Yes	No	1. High expression of NDRG2 reduced phospho-AKT and induced phosphorylation of p38 MAP kinase.
							2. T47D and MCF7 cells (less malignant) had strong expressions of NDRG2, but NDRG2 was not detected in MDA-MB-453 and MDA-MB-231 cells (highly malignant).
		Shon, 2009 [66]	MDA-MB-231, MCF-7	Not reported	Yes	No	1. NDRG2 induced BMP-4 and suppressed MMP-9 activity.
							2. NDRG2 expression inhibited the in vitro migration and invasion potential of breast cancer cells.
		Zheng, 2010 [77]	MCF-7, Bcap-37	Not reported	Yes	No	1. NDRG2 suppressed adhesion and invasion of breast cancer cells.
		Lorentzen, 2011 [69]	Not reported	48	Not reported	Not reported	1. *NDRG2* mRNA expression was reduced in breast cancer relative to normal tissue.
		Oh, 2012 [68]	4T1	189	Yes	Yes	1. High levels of *NDRG2* correlated with better disease-free survival but not with overall survival.
							2. NDRG2 overexpression reduced migration and invasion in vitro.
							3. High levels of NDRG2 reduced tumor growth in vivo.
		Kim, 2014 [73]	MDA-MB-231	Not reported	Yes	No	1. Overexpression of NDRG2 induced apoptosis.
		Kim, 2014 [71]	MDA-MB-231	Not reported	Yes	No	1. Overexpression of NDRG2 inhibited the epithelial–mesenchymal transition through STAT3/Snail signaling.
		Kim, 2014 [72]	MDA-MB-231, MCF-7	Not reported			1. Overexpression of NDRG2 downregulated COX-2 through NF-kB signaling.
							2. Overexpression of NDRG2 reduced migration and invasion of MDA-MB-231 cells.
		Kim, 2016 [74]	4T1	Not reported	Yes	No	1. *NDRG2* expression in breast cancer cells inhibited osteoclast differentiation.
		Wei, 2017 [65]	MCF-7, MDA-MB-231, T47D	Not reported	Yes	No	1. Doxorubicin-resistant breast cancer cells expressed reduced levels of NDRG2.
		Lee, 2021 [70]	MDA-231, MCF-7, 4T1		Yes	No	1. NDRG2 negatively regulated PDL1 expression in malignant breast cancer cells by suppressing NF-kB signaling.
							2. NDRG2 expression was inversely correlated with PDL1 expression, mainly in TNBC.
		Zhai, 2022 [64]	MDA-231, SK-BR-3, HCC2157, BT474, HCC1569, T47D	120	Yes	Yes ^[g]^	^[g]^ In vivo studies focused on miR-181a-5p, not NDRG2.
							1. *NDRG2* expression was high in normal tissue compared with breast cancer.
							2. Patients with *NDRG2* -low tumors had worse outcomes than those with *NDRG2* -high tumors.
							3. MiR-181a-5p inhibited NDRG2 to promote proliferation and invasion via activation of the PTEN/AKT pathway.
	**Tumor** **promoter**	Kloten, 2016 [75]	HCC1806, BT20, MCF-7	62 + 211 ^[h]^	Yes	No	^[h]^ Involved 62 tissue samples, 45 from breast tumors and 17 from adjacent normal tissues, and tissue microarray with 211 patient samples.
							1. Basal-like tumors had abundant *NDRG2* expression compared with luminal tumors.
							2. Basal-like tumors had positive correlation with *NDRG2* expression.
							3. Tumor suppressor function could be limited to luminal and basal-B subtypes, but NDRG2 acted as a tumor promoter in basal-A subtype.
**NDRG3**	**Tumor suppressor**	Estiar, 2017 [79]	Not reported	88	Not reported	Not reported	1. NDRG3 was downregulated in patients with breast cancer, particularly those with advanced disease.
	**Tumor** **promoter**	Kim, 2019 [78]	Not reported	1339	Not reported	Not reported	1. Patients with *NDRG3*-positive invasive breast cancer had worse overall survival than those with NDRG3-negative tumors.
							2. *NDRG3* independently predicted worse overall survival and disease-free survival.
**NDRG4**	**Tumor** **suppressor**	Jandrey, 2019 [80]	MCF-7, T47D, MDA-MB-231, MDA-MB-435	61	Yes	No	1. *NDRG4* was highly methylated in breast cancer samples relative to normal breast.
							2. Patients with *NDRG4*-methylated tumors had worse overall survival and distant metastasis-free survival.
							3. *NDRG4* methylation status was an independent predictor of distant metastasis-free survival.

## 4. NDRG Signaling Pathways

The signaling pathways involved in the regulation of NDRGs in various types of cancer have been reviewed elsewhere [23,81,82]. NDRG1 has been shown to suppress tumor growth, migration, and invasion by suppressing several known pathways, including NF-κB, E-cadherin, EGFR, and WNT/β-catenin [45,82,83]. More recently, the mTOR/AKT pathway was implicated in the regulation of NDRG1 as a tumor/metastasis promoter [51], and TGFβ has also been found to be responsible for NDRG1 having tumor-promoting activity via regulation of the epithelial–mesenchymal transition, metastasis, and cancer stem cells [63] (Figure 3).

In contrast to NDRG1, NDRG2 acts as a tumor suppressor, inhibiting proliferation and cell survival via regulations of MAPK/STAT3, cyclin-D1, β-catenin, NF-κB, E-cadherin, and other signaling pathways [23,81,84,85]. Thus far, no pathway has been identified as supporting a role for NDRG3 being either a breast tumor suppressor or promoter. NDRG4 has been shown to act via signaling pathways such as cyclin-D1, p27, XIAP, and survivin to increase tumor growth and inhibit apoptosis in other types of cancer [23], but no information on NDRG4 signaling pathways has been reported in breast cancer.

## 5. NDRGs and Amplification in Breast Cancer

One of the hallmarks of cancer is genomic instability and mutation, some of which result in gene amplification, which is defined as an increase in the copy number of certain regions of the chromosome; amplicon refers to the region that was amplified [86]. The amplification of certain genes can benefit cancer cells by increasing their proliferation and enhancing drug resistance [87]. Oncogene amplification is also correlated with the aggressiveness of cancer cells and poor prognosis for cancer patients. Oncogenes that are well-known to be amplified in cancer include *MYC*, *EGFR*, *ERBB2*, *CCND1*, and *MDM2* [86,88,89]. One analysis of primary breast cancer samples revealed that *ERBB2, FGFR1, MYC, CCND1*, and *PIK3CA* were commonly amplified, and other well-known oncogenes such as *CCND2, EGFR, FGFR2*, and *NOTCH3* were amplified at lower frequencies [88].

*NDRG1* is located on chromosome 8q24.3, near *MYC*, and amplification of this region is both common and prognostic in breast cancer [90,91]. Our own analysis of 816 breast tumors from The Cancer Genome Atlas (TCGA) has revealed that *NDRG1* is one of the six most commonly amplified genes in breast cancer, which include *MYC*, *RAD21*, *EXT1*, *NDRG1*, *UBR5*, *CCND1* (Figure 4A) [92]; *NDRG1* was found to be amplified in 17% of patients, *NDRG2* in 0.5%, *NDRG3* in 2.9%, and *NDRG4* in 0.9% (Figure 4B). We have confirmed these results by analyzing METABRIC, an independent dataset comprising more than 2000 breast cancer patient samples. Our analysis revealed that *NDRG1* is among the most frequently amplified genes in this cohort of breast cancer patients (Figure 4A). Specifically, we observed *NDRG1* amplification in 23% of breast cancer samples, whereas *NDRG2* was amplified in 0.9% cases, *NDRG3* in 2.5% and *NDRG4* in 0.3% (Figure 4B) [93,94,95]. However, the presence of gene amplification is not a guarantee of overexpression [96,97]. In fact, one study of cervical cancer revealed that some amplified genes showed no changes in expression, and others were repressed [96]. Even in breast cancer, wherein HER2 amplification is historically correlated with overexpressions of mRNA and protein [98,99], Luoh et al. have shown that a subset of patients with HER2 gene amplification did not overexpress the HER2 protein, which may affect the response to HER2-targeted therapies [100]. For this reason, even though we observed a high amplification of *NDRG1* in the datasets we analyzed, we have confirmed that the amplification correlated with higher levels of both *NDRG1* mRNA and protein (*p* < 0.0001; Figure 4C). Moreover, patients with *NDRG1* amplification had worse overall survival than patients with diploid *NDRG1* (*p* ≤ 0.01; Figure 4D).

## 6. NDRGs and Outcomes in Breast Cancer

Our group has previously analyzed several independent cohorts of breast cancer patients and shown that NDRG1 was expressed at higher levels in tumor samples than in normal tissue and was more highly expressed in ER^−^ tumors than in ER^+^ tumors. Regarding molecular subtype, *NDRG1* expression was highest in more aggressive subtypes (HER2^+^ and basal-like), and its expression was associated with a higher pathological grade. Patients with *NDRG1*-high tumors also had worse overall and metastasis-free survival [49]. For each dataset, patients were stratified as high or low according to their median *NDRG1-4* expression within that dataset. Kaplan–Meier curves and log-rank tests were used to compare survival distributions. Mann–Whitney U tests were used to compare two groups and one-way analysis of variance was used for multiple experimental groups. The analysis was performed using GraphPad software (GraphPad Prism 9, La Jolla, CA, USA). An analysis of additional independent cohorts [92,93,94,95,101,102] has confirmed that *NDRG1* expression is higher in ER^−^ tumors (*p* < 0.0001; Figure 5A) and in basal-like tumors (*p* ≤ 0.05; Figure 5B). Moreover, in two independent cohorts, *NDRG1* status correlated with worse overall survival, wherein patients with *NDRG1*-high tumors had worse outcomes (*p* ≤ 0.02; Figure 5C).

In contrast to *NDRG1*, we found that *NDRG2* expression was higher in normal samples than in tumor samples (*p* < 0.0001; Figure 6A). However, like *NDRG1*, patients with aggressive tumor types such as ER– and basal-like tumors expressed higher levels of *NDRG2* relative to those with ER+ tumors and non-basal subtypes (*p* < 0.0001; Figure 6B,C). Analyses of *NDRG2* expression according to tumor grade showed no difference between grades 1 and 3, but grades 2 and 3 differed in the Hatzis dataset (*p* = 0.0011), and no differences were observed in the Desmedt cohort (Figure 6D), highlighting a lack of consistency between the datasets. Moreover, *NDRG2* expression levels (high vs. low) remained unaffected by the metastasis status of tumors and did not affect the overall survival or metastasis-free survival in any of the datasets analyzed (Figure 6E–G) [92,103,104,105].

Analyses of *NDRG3* expression showed no significant differences in expression in normal vs. breast cancer tissues (Figure 7A). In terms of ER status, ER^−^ tumors had higher expressions of *NDRG3* than ER+ tumors (*p* = 0.0261) in the TCGA dataset, but no difference was found in the Hatzis dataset (Figure 7B). NDRG3 was expressed at similar levels among the molecular subtypes, being higher in luminal B than in basal-like subtypes (*p* ≤ 0.02, Figure 7C). *NDRG3* expression levels were not correlated with pathological grade (Figure 7D). Similar to ER status, only the TCGA dataset showed a significant difference in overall survival; that is, *NDRG3*-high patients had worse outcomes than *NDRG3*-low (*p* = 0.0457). However, the Desmedt dataset showed no such significant differences (Figure 7E). Moreover, no difference was found in *NDRG3* expressions according to metastasis or no metastasis (Figure 7F), and *NDGR3* expression did not affect metastasis-free survival (Figure 7G) [92,103,104,105].

Finally, *NDRG4* expression was similar in normal and breast tumor tissues (Figure 8A) and was higher in ER^−^ tumors than in ER^+^ tumors (*p* ≤ 0.003, Figure 8B). Basal-like tumors showed higher expressions of *NDRG4* than the luminal A and B subtypes (*p* < 0.004), but NDRG4 expressions did not differ between the two aggressive subtypes (HER2+ and basal-like) (Figure 8C). Similarly, *NDRG4* expressions did not differ by tumor grade (Figure 8D) and were not associated with overall survival (Figure 8E). *NDRG4* expressions did differ by metastasis vs. no metastasis (*p* = 0.0159, Figure 8F), and *NDRG4*-high patients had worse metastasis-free survival (*p* = 0.0005, Figure 8G), but only in the Hatzis dataset [92,103,104,105].

## 7. Conclusions

Our comprehensive literature review that encompasses multiple lines of evidence and experimental data regarding the role of the NDRGs in breast cancer has revealed the following key findings. First, NDRG1 seems to have a dualistic function in breast cancer, acting as both a tumor/metastasis promoter and suppressor, but mounting evidence supports a pro-metastasis role in aggressive breast cancers. NDRG2 has been described mostly as a tumor suppressor, and more information is needed regarding the functions of NDRG3 and NDRG4 in breast cancer. NDRGs have been shown to be involved in different signaling pathways and to have different cellular functions in breast cancer. A further characterization of NDRGs is warranted to best tap into their potential as therapeutic targets in breast cancers.

## Figures and Tables

**Figure 1 cancers-16-01342-f001:**
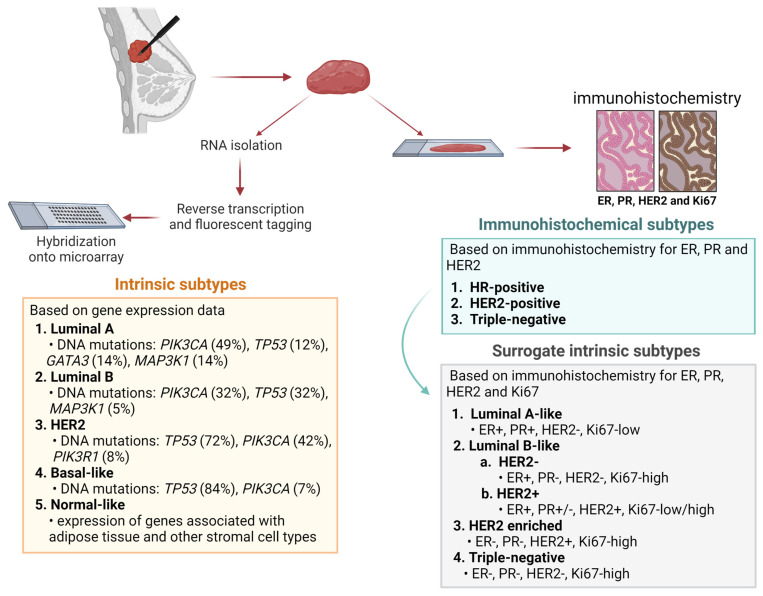
Breast cancer classification. Breast cancer can be classified according to gene expression data (intrinsic subtypes) or immunohistochemical staining (surrogate intrinsic subtypes). The intrinsic subtypes classify breast cancer based on the 50-gene expression signature PAM50. The immunohistochemical classification was initially based on the protein expression of the estrogen receptor (ER), progesterone receptor (PR), and the human epidermal growth factor (HER2). Hormone receptor (HR)-positive refers to tumors that express ER and/or PR, whereas triple-negative tumors do not express ER, PR, or HER2. Subsequent addition of the proliferation marker Ki67 led to creation of the surrogate intrinsic subtypes, which are currently used in the clinic as the basis for treatment decisions. Created with BioRender.com.

**Figure 2 cancers-16-01342-f002:**
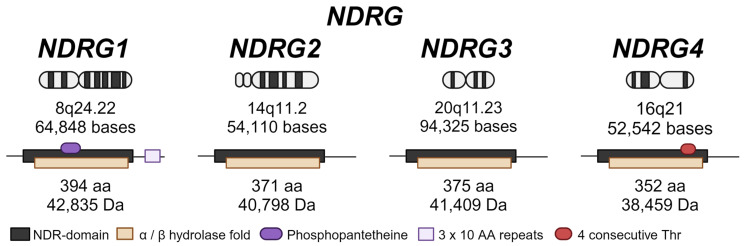
NDRG family members, including their chromosome localization and protein structure. Created with BioRender.com.

**Figure 3 cancers-16-01342-f003:**
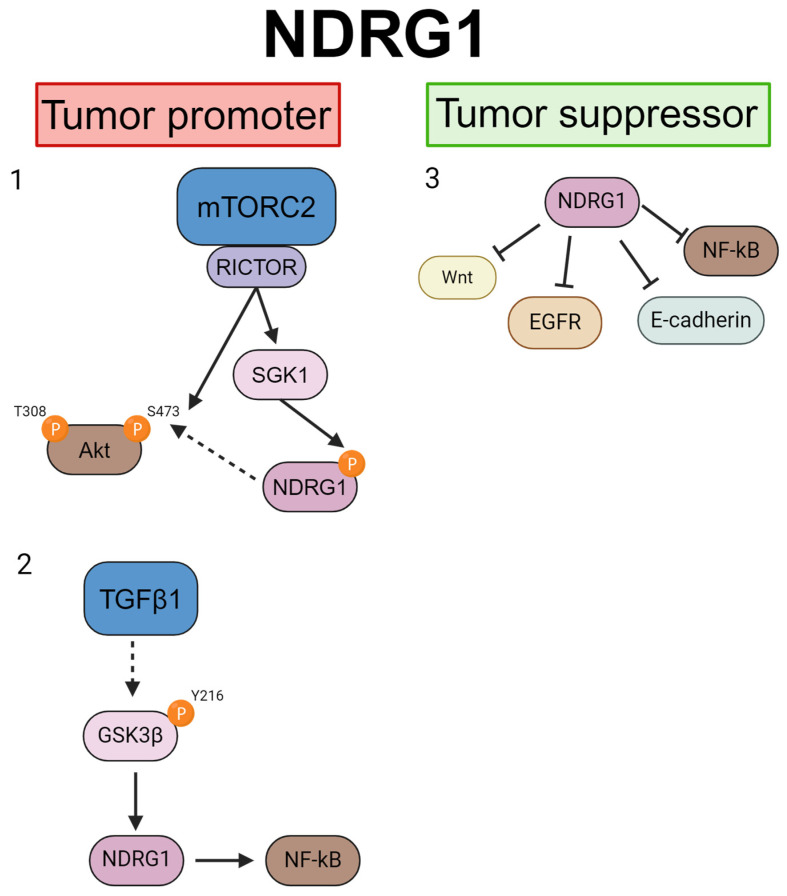
NDRG1 pathways in breast cancer. NDRG1 can promote tumor growth via (1) mTORC2/AKT pathway [51] or (2) TGFβ1 [63]; on the other hand, several NDRG1-regulated pathways (3) have been reported to suppress tumor growth [45,82,83]. Created with BioRender.com.

**Figure 4 cancers-16-01342-f004:**
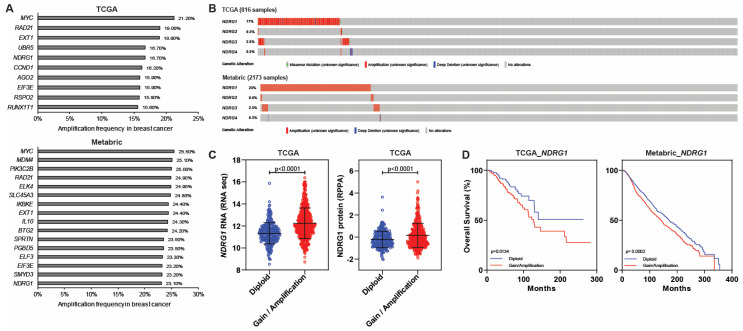
*NDRG1* amplification in breast cancer. (**A**) Top genes amplified in breast cancer. (**B**) Frequency of amplification of NDRGs in breast cancer. (**C**) *NDRG1* mRNA and protein levels plotted with copy number alterations. (**D**) Overall survival analysis for patients whose tumors show *NDRG1* gain or amplification versus tumors that are diploid.

**Figure 5 cancers-16-01342-f005:**
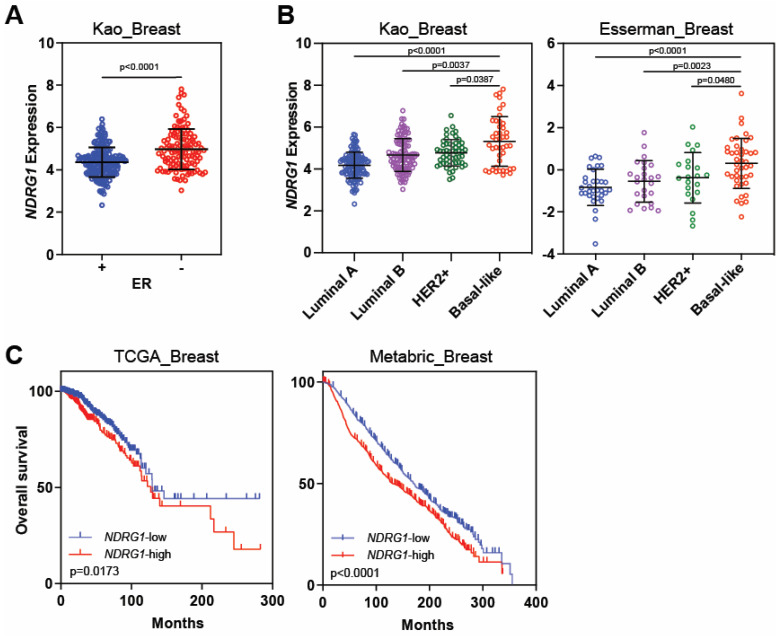
*NDRG1* expression in breast cancer. (**A**) *NDRG1* expression according to estrogen receptor (ER) status. (**B**) *NDRG1* expression by molecular subtype of breast cancer. Black lines in each group indicate medians ± SD. *p* values were calculated with 2-sided Mann–Whitney U tests. (**C**) Kaplan–Meier curves for overall survival according to *NDRG1* expression; *p* values were obtained with log-rank tests.

**Figure 6 cancers-16-01342-f006:**
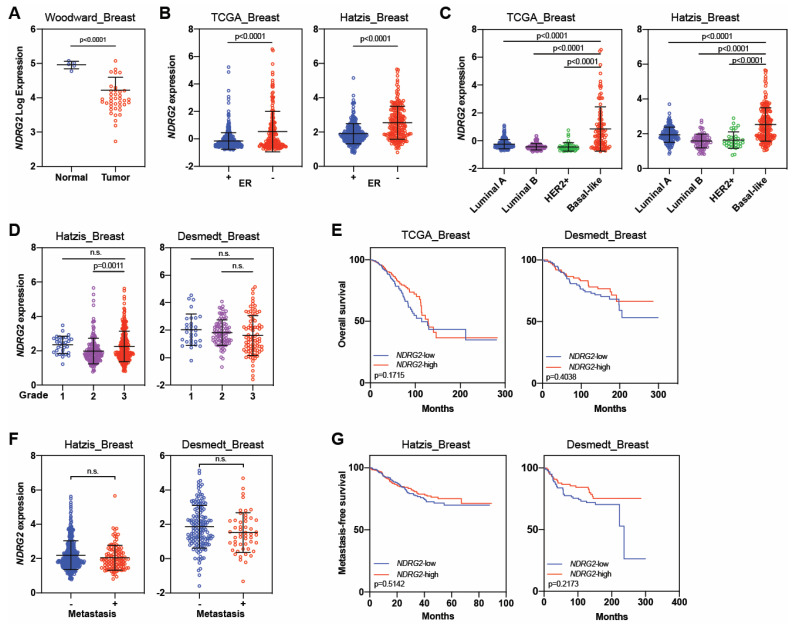
*NDRG2* expression in breast cancer. (**A**) *NDRG2* expression in normal tissues versus breast tumor samples. (**B**) *NDRG2* expression in patients with estrogen receptor-positive (ER^+^) and ER-negative (ER^−^) tumors. (**C**) *NDRG2* expression by molecular subtype of breast cancer. (**D**) *NDRG2* expression by pathological tumor grade. (**E**) Kaplan–Meier curves for overall survival according to *NDRG2* expression. (**F**) *NDRG2* expression by metastasis or no metastasis. (**G**) Kaplan–Meier curves for metastasis-free survival according to *NDRG2* expression. Independent cohorts were used to analyze *NDRG2* expression and survival outcomes. Black lines in each group indicate medians ± SD. *p* values were calculated with 2-sided Mann–Whitney tests (**A**–**D**,**F**) or log-rank tests (**E**,**G**). n.s. = not significant.

**Figure 7 cancers-16-01342-f007:**
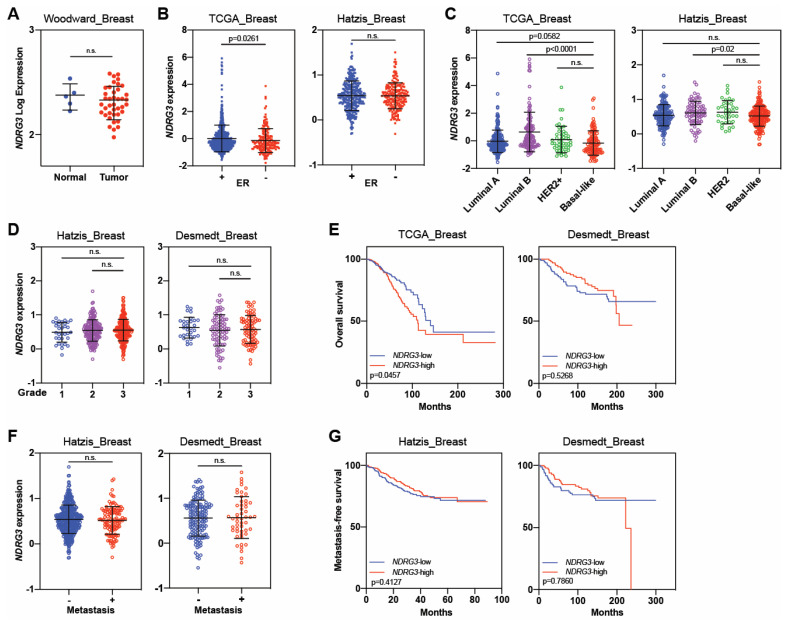
*NDRG3* expression does not influence outcome in patients with breast cancer. (**A**) *NDRG3* expression in normal versus breast cancer tissue samples. (**B**) *NDRG3* expression in patients with estrogen receptor-positive (ER+) and ER-negative (ER–) tumors. (**C**) *NDRG3* expression by molecular subtype of breast cancer. (**D**) *NDRG3* expression by pathological tumor grade. (**E**) Kaplan–Meier analysis for overall survival by *NDRG3* expression. (**F**) *NDRG3* expression by metastasis or no metastasis. (**G**) Kaplan–Meier analysis of metastasis-free survival analysis by *NDRG3* expression. Independent cohorts were used to analyze *NDRG3* expression and survival outcomes. Black lines in each group indicate medians ± SD. *p* values were calculated with Mann–Whitney U test (**A**–**D**,**F**), and log-rank test (**E**,**G**). n.s. = not significant.

**Figure 8 cancers-16-01342-f008:**
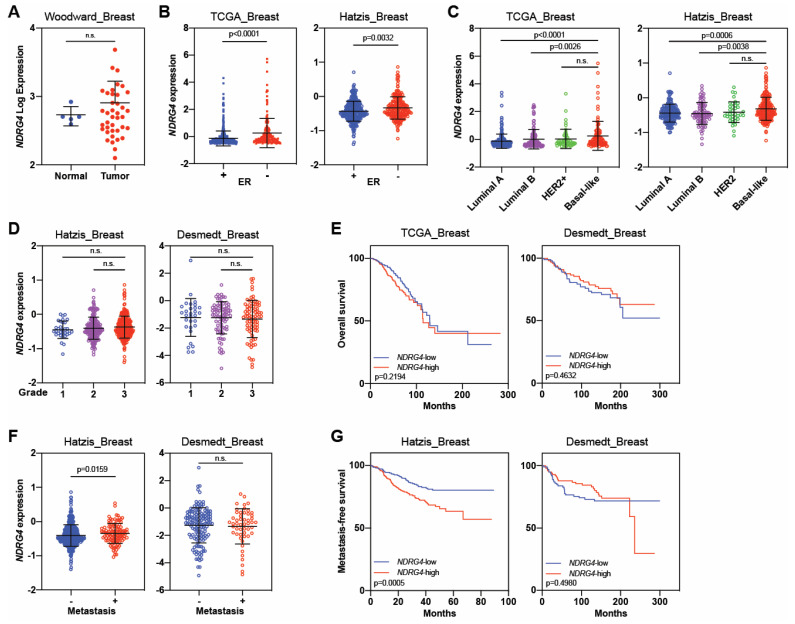
*NDRG4* is overexpressed in aggressive breast cancer. (**A**) *NDRG4* expression in normal tissues versus breast tumor samples. (**B**) *NDRG4* expression in patients with estrogen receptor-positive (ER+) and ER-negative (ER–) tumors. (**C**) *NDRG4* expression by molecular subtype of breast cancer. (**D**) *NDRG4* expression by pathological tumor grade. (**E**) Kaplan–Meier curves for overall survival according to NDRG4 expression. (**F**) *NDRG4* expression by metastasis or no metastasis. (**G**) Kaplan–Meier curves for metastasis-free survival according to *NDRG4* expression. Independent cohorts were used to analyze *NDRG4* expression and survival outcomes. Black lines in each group indicate medians ± SD. *p* values were calculated with Mann–Whitney U tests (**A**–**D**,**F**) or log-rank tests (**E**,**G**). n.s. = not significant.

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
