# Peer review of "NDRGs in Breast Cancer: A Review and In Silico Analysis"

_cancers, 2024, doi:10.3390/cancers16071342_

Round 1
Reviewer 1 Report
Comments and Suggestions for Authors
This is an excellent review of a field that has not been review in at least 5+ years from what I can find. The content is current, informative and allows the reader to be informed about the role of various NDRGs in biology and specifically breast cancer. A lot of focus has been on NDRG1 so it is nice to see that the authors have discussed all family members. A few recommendations:
1. Please spell out NDRG in the "simple summary" section;
2. I think the "abstract" need a bit more length with more broad statements about NDRGs;
3. For Line 68-70 statement about NDRGs in cancer...it think its informative if the authors can state 3-4 sentences about NDRG in other prominent cancers so the reader does not have to search the literature to find out this information;
4. For section 2 "NDRG family proteins", are there other locations for NDRGs other than cytosol? Any other tissue specificities as only NDRG3 is mentioned?
5. Any post-translational modifications for NDRGs? Please add to section 2.
6. I commend you on the summary in the table 1.
7. It seems that Supplementary figures 1 and 2 should be in the main list of figures? If you do not have room for it due to page restriction I do understand that.
Excellent paper.
6.
Reviewer 2 Report
Comments and Suggestions for Authors
The authors have reviewed the role played by N-myc downstream regulated gene (NDRG) family members in tumor promoters as well as tumor suppressors. They have done meta-analyses of the transcriptomic data available in the public domain. Review is well-research and has clarity. Few improvements are suggested below:
1. A new figure comparatively summarizing all the four human NDRG proteins must be added. Categories can be chromosomal location, sequence length, MW, size, 3D structure, major function, tissue expression, pathways involved in and cancer association.
2. Alphabetical footnotes for Table 1 are missing
3. Schematic new figure should be added especially for NDRG1 and NDRG2 canonical signaling pathways with respect to breast cancer to facilitate better understanding of their role holistically.
